# Validating agent-based simulation model of hospital-associated *Clostridioides difficile* infection using primary hospital data

**Elizabeth Scaria[1], Nasia Safdar[2,3,4], Oguzhan Alagoz[1,2]***

**1** Department of Industrial and Systems Engineering, University of Wisconsin- Madison, Madison, WI, United States of America, **2** Population Health Sciences, School of Medicine and Public Health, University of Wisconsin- Madison, Madison, WI, United States of America, **3** Division of Infectious Diseases, Department of Medicine, University of Wisconsin-Madison, School of Medicine and Public Health, Madison, WI, United States of America, **4** William S. Middleton Memorial Veterans Hospital, Madison, WI, United States of Ameirca

* alagoz@engr.wisc.edu

**Data Availability Statement:** All relevant data are within the manuscript and its Supporting Information files.

## Abstract

As agent-based models (ABMs) are increasingly used for modeling infectious diseases, model validation is becoming more crucial. In this study, we present an alternate approach to validating hospital ABMs that focuses on replicating hospital-specific conditions and proposes a new metric for validating the social-environmental network structure of ABMs. We adapted an established ABM representing *Clostridioides difficile* infection (CDI) spread in a generic hospital to a 426-bed Midwestern academic hospital. We incorporated hospital-specific layout, agent behaviors, and input parameters estimated from primary hospital data into the model, referred to as H-ABM. We compared the predicted CDI rate against the observed rate from 2013–2018. We used colonization pressure, a measure of nearby infectious agents, to validate the socio-environmental agent networks in the ABM. Finally, we conducted additional experiments to compare the performance of individual infection control interventions in the H-ABM and the generic model. We find that the H-ABM is able to replicate CDI trends during 2013–2018, including a roughly 46% drop during a period of greater infection control investment. High CDI burden in socio-environmental networks was associated with a significantly increased risk of *C. difficile* colonization or infection (Risk ratio: 1.37; 95% CI: [1.17, 1.59]). Finally, we found that several high-impact infection control interventions have diminished impact in the H-ABM. This study presents an alternate approach to validation of ABMs when large-scale calibration is not appropriate for specific settings and proposes a new metric for validating socio-environmental network structure of ABMs. Our findings also demonstrate the utility of hospital-specific modeling.

## Introduction

Agent-based simulation models (ABMs) have been increasingly used to model infectious diseases. Although ABMs may overcome major limitations of conventional modeling approaches

**Funding:** Financial support for this study was provided entirely by a grant from the National Institute Of Allergy And Infectious Diseases of the National Institutes of Health under Award Number DP2AI144244 awarded to NS (https://www.niaid.nih.gov/). The funders had no role in study design, data collection and analysis, decision to publish, or preparation of the manuscript.

**Competing interests:** I have read the journal's policy and the authors of this manuscript have the following competing interests: Oguzhan Alagoz has served as a consultant to Johnson & Johnson, Bristol Myers Squibb, Exact Sciences, and Biovector Inc. all of which are outside of the submitted work. Other authors report no conflict of interest. This does not alter our adherence to PLOS ONE policies on sharing data and materials.

**Abbreviations:** CDI, *Clostridioides difficile* infection; ABM, Agent-based model; H-ABM, Hospital agent-based model; MCP, Mean colonization pressure; HA-CDI, Hospital-associated CDI; CA-CDI, Community-associated CDI.

such as state-transition models and discrete-event simulation, their applicability is limited because of their extensive data needs [1]. ABMs need large amounts of data not only for estimating model parameters such as those related to agent interactions, but also for conducting validation.

The International Society for Pharmacoeconomics and Outcomes Research-Society for Medical Decision Making (ISPOR-SMDM) Modeling Good Research Practices Task Force notes the success of healthcare models depends on trust and confidence, which could be achieved via transparency and validation [2]. However, creating and validating complex ABMs can be especially difficult because of limited available primary historical data. While several ABMs of healthcare settings exist, few have been validated using primary data from an extant system: often, validation efforts use community- or national-level historical data [3–6]. When primary historical data is used in the model validation process for ABMs of healthcare settings, it is typically used to calibrate certain model parameters and therefore does not serve as an external validator [5, 7]. Unvalidated, generic models may not be able to fully represent a specific hospital's characteristics and therefore may be of diminished utility in decision making. Challenges associated with validation posed by insufficient data are amplified when modeling specific hospitals because primary hospital data can be particularly scarce given the high cost of data collection. This lack of data is exemplified by the number of hospital ABMs that leverage at least some mix of primary hospital data and literature or secondary data sources [5, 8–11]. To the best of our knowledge, there is currently a dearth of formalized procedures that explore possibilities for blending literature and primary data sources to contribute to model validation.

Additionally, ABMs depend heavily on the networks facilitating interactions between agents and their environment, yet validating socio-environmental networks is especially challenging [12]. For example, ABMs may record the frequency of interactions among agents in a system (e.g., patients to patients, or patients to healthcare workers) that contribute to the transmission of a disease, but there is paucity of data and methods to validate these frequencies and the overall socio-environmental network structure. As a result, modelers might need to consider indirect metrics of network behavior to validate socio-environmental networks, such as emergent network effects, but to the best of our knowledge this approach has not been employed in the context of modeling disease transmission in a hospital setting.

To this end, this study demonstrates how an ABM and its associated socio-environmental network structure could be externally validated using primary hospital data and novel risk factors. For this purpose, we use a previously developed ABM of hospital-associated *Clostridioides difficile* infection (CDI), a gastrointestinal infection associated with abdominal pain, diarrhea, and death in severe cases. CDI is one of the most common hospital associated infectious diseases in the United States, with up to 500,000 cases and 29,000 attributed deaths per year, and its control has long been a national public health priority [13]. Using specific characteristics from a 426-bed academic hospital in the Midwestern United States, we adapted the previously developed generic model to the target academic hospital-specific setting. To the best of our knowledge, previously published ABMs of hospital-associated CDI have neither been externally validated with historical data, nor have they been tailored to fit a specific hospital. Adapting a generic ABM to a specific hospital environment is not an insignificant task, as thorough validation requires investment in rigorous data gathering, and the benefit of this undertaking is unclear. Thus, to assess the benefit of hospital-specific modeling, we compared the relative effectiveness of different infection control interventions across the generic and hospital-specific models. These experiments are intended to demonstrate the value of hospital-specific modeling, and support arguments in favor of greater deliberation in model building and validation.

## Methods

### Background

CDI is one of the most common healthcare-associated infections in the United States and primarily spreads via the fecal-oral route through infectious spores shed in the stool of infected patients [13, 14]. Removal of these spores from hard surfaces or hands requires the use of strong sporicidal agents or vigorous soap and water cleansing, respectively. CDI susceptibility can vary, with risk factors including advanced age, immunosuppression, and antibiotic exposure [15]. Healthcare facilities differentiate between CDIs likely developed in the hospital or the community, referring to cases as hospital-associated CDI (HA-CDIs) or community-associated CDI (CA-CDI), respectively [16].

To limit the prevalence of CDI in hospitals, infection control programs develop and deploy interventions, with varying costs and effectiveness, aimed at eliminating *C. difficile* spores. Infectious disease control guidelines recommend several interventions, including increased hand hygiene, environmental cleaning, and testing at patient admission, among others [14].

### Overview of the existing agent-based simulation model developed for a generic hospital setting

Barker, et al and Codella, et al previously developed an ABM of CDI spread in a generic hospital [5, 8]. The ABM includes four types of agents: patients, nurses, doctors, and visitors. Patients interact with the environment and other agents during their stay, and each interaction may propagate infection with a probability dependent on the infection control interventions in place. All patients that test positive for CDI start a 14-day course of antibiotics and extend their length of stay by approximately 2.3 days [17, 18]. At the end of a stay, the patient is discharged, and the agent is removed from the model. In accordance with typical hospital protocols, patients with known CDI infections may be discharged even if they have not completed their course of antibiotics. Section A in S1 Appendix contains behavior logic flow diagrams for all agent types in the ABM.

In this model, patients are the only agents that may develop CDI, while healthcare worker and visitor agents may become transiently contaminated [5, 8]. Patient CDI state is modeled using a discrete-time Markov chain updated every 6 hours in-simulation with nine states: susceptible, exposed, colonized, infected, recurrent colonized, recurrent infected, cleared, dead, and non-susceptible [8]. Patients may only transition into the "exposed" state if they encounter *C. difficile* spores. Patients may only transition from the "non-susceptible" to "susceptible" states if they begin a course of a high-risk antibiotics. All other disease state transitions are determined using a transition probability matrix, which was calibrated using long term probabilities estimated from literature and hospital data [5]. Fig 1 visualizes this discrete-time Markov chain, with the dashed arrows representing infection control dependent transitions.

To reduce variability and compare results across different conditions, the model uses common random numbers generated by the Colt Project's Mersenne Twister algorithm [19]. The ABM was developed in the Java programming language using Microsoft's Visual Studio editor. Each set of 5000 replications of the ABM required roughly one hour on a single desktop computer with an Intel Core i5-8500 CPU and 16GB RAM. Further details of the generic ABM are available in previous publications [5, 8, 20].

### Adapting the generic hospital ABM to target hospital setting

The H-ABM simulates CDI-related events in 426 single occupant patient rooms of a Midwestern academic hospital. To better model the target hospital, we adapted the generic hospital

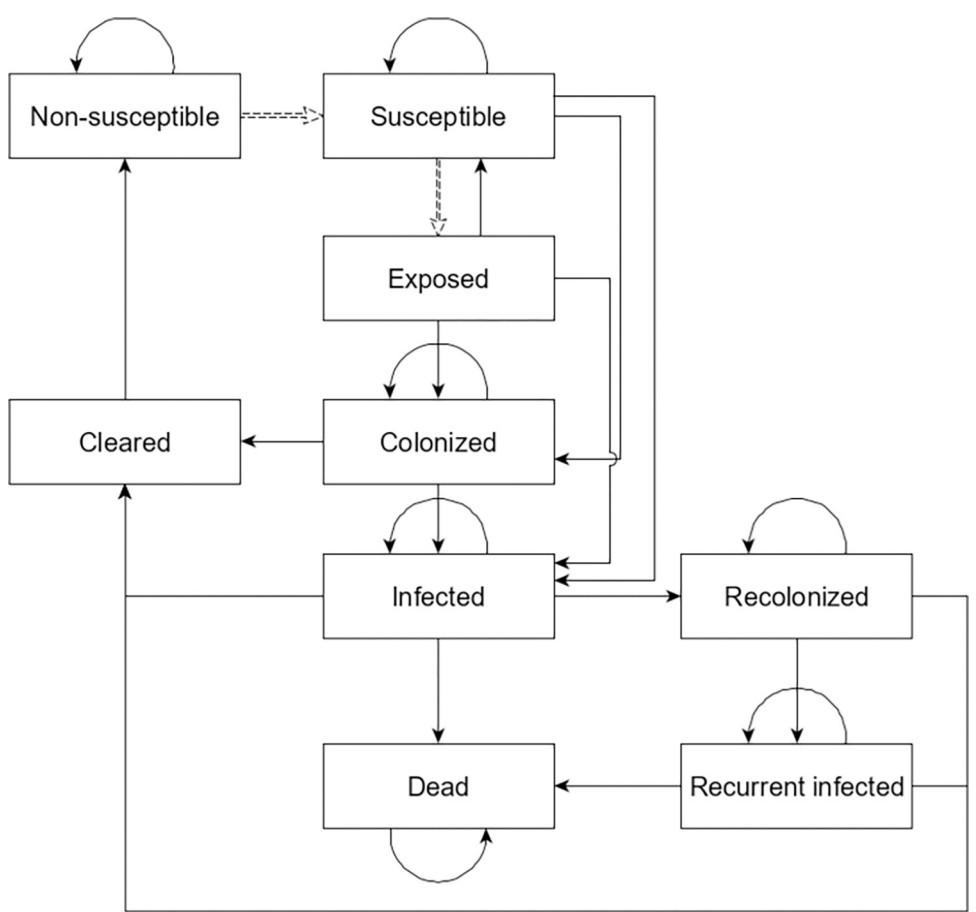

**Fig 1. Discrete-time Markov chain structure representing the progression of CDI in an individual patient.**

ABM to include several hospital-specific features. Patient rooms are organized into 18 wards of varying size according to the target hospital layout. We included an additional space representing the hospital cafeteria, which healthcare workers may visit once per shift when not interacting with patients. The H-ABM also includes common room spaces for patients/visitors and healthcare workers within each ward. In this model, we assume that nurses see patients within a single ward and use only the common rooms in that ward, while doctors consult for patients across the hospital and use common rooms across multiple wards. Fig 2 illustrates the modeled layout of the target hospital.

We focused on replicating CDI-related trends between years 2013 and 2018 for which the most complete data on interventions and infections are available, as described in Section B of S1 Appendix. Section C in S1 Appendix provides a detailed description of the major changes between H-ABM and the ABM representing a generic hospital. Tables C1 and C2 in S1 Appendix provide further parameters governing agent behavior and environmental conditions in H-ABM.

## Modeling infection control interventions

The H-ABM included several infection control interventions: nurse and doctor hand hygiene, nurse and doctor contact precautions, daily environmental cleaning, terminal environmental cleaning, visitor hand hygiene, visitor contact precautions, patient hand hygiene, patient

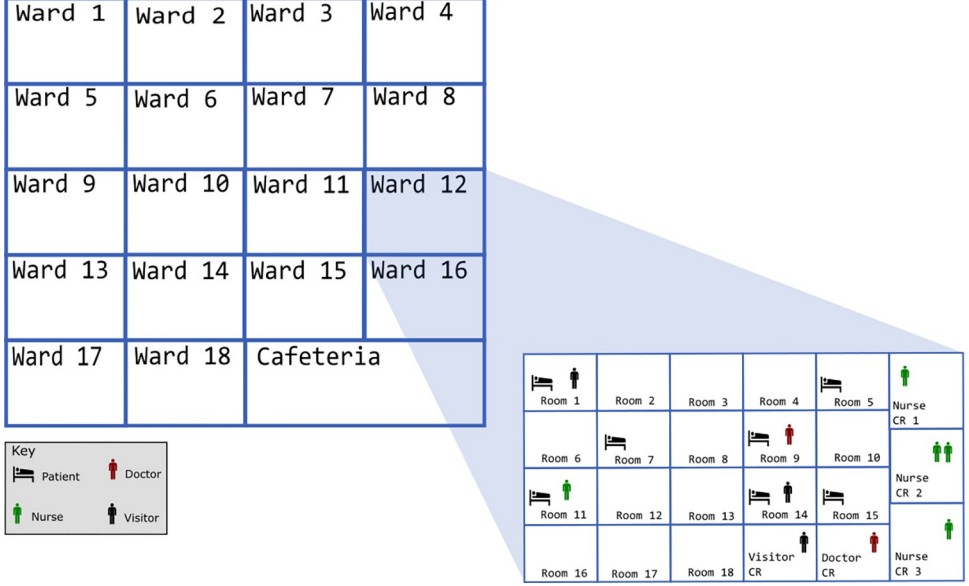

**Fig 2. Layout of the hospital simulated in H-ABM.**

transfer, and *C. difficile* screening at admission. To simulate the HA-CDI rate for the years 2013–2018, compliance and effectiveness were estimated for each intervention. We defined "compliance" as the proportion of agents following infection control protocol in a single opportunity, and "effectiveness" as the probability that each infection control event successfully eliminates or blocks transmission of *C. difficile*. Several parameter estimates varied for CDI and non-CDI patient rooms. Further details of infection control interventions are provided in Section D of S1 Appendix.

### Parameter estimation and data sources

Several parameters in our model were estimated from primary hospital data. To estimate the proportion of susceptible patients by ward, we used hospital admission data to determine the number of patients aged 65 years or older, or who had been taking antibiotics upon admission. We also assumed that all patients admitted to wards specialized in caring for high-risk classes of patients (e.g., solid organ transplant) are susceptible to CDI. High risk wards were identified by a clinical expert on CDI in the target hospital. We also used hospital administrative data to estimate the daily patient arrival rate, antibiotic use by ward, and average base length of stay by ward. The proportion of susceptible patients and the antibiotic use in the target hospital were assessed for each of the years 2013–2018 to simulate broader trends in antibiotic use during this period in the US [21, 22]. A full description of hospital layout and parameters are included in Table 1, Tables C1-C4 in S1 Appendix, and Fig 2. Parameters related to CA-CDI, such as the proportion of admitted patients that are colonized or infected, were estimated from literature. Parameters unlikely to vary significantly from the previously published generic ABM, such as the transfer efficiency of *C. difficile* between surfaces, were not changed for the hospital-specific model.

Another crucial input for representing CDI-related events is the use of infection control interventions, which have varying degrees of fidelity over time. We estimated parameters related to interventions using hospital data of monthly observed healthcare worker hand hygiene events from 2013–2018. This data included counts of nurses and doctors performing hand hygiene actions while seeing patients, and the number of hand hygiene events using soap

**Table 1. Hospital and infection control intervention parameter estimates, by year.**

| Intervention | Year | | | | | | Source |
|---|---|---|---|---|---|---|---|
| | 2013 | 2014 | 2015 | 2016 | 2017 | 2018 | |
| Hand Hygiene Parameters | | | | | | | |
| Standard Patient | | | | | | | |
| Healthcare workers compliance with hand hygiene (effectiveness) | 67% (34%) | 68% (33%) | 70% (34%) | 80% (34%) | 83% (35%) | 84% (35%) | [23–27] (HD) |
| Visitor compliance with hand hygiene (effectiveness) | 37% (34%) | 37% (33%) | 38% (34%) | 61% (34%) | 69% (35%) | 69% (35%) | [23, 24, 28] ([13]) |
| Patient compliance with hand hygiene (effectiveness) | 27% (34%) | 27% (33%) | 28% (33%) | 28% (34%) | 28% (35%) | 28% (35%) | [29–31] ([13]) |
| CDI Patient | | | | | | | |
| Healthcare workers compliance with hand hygiene (effectiveness) | 86% (84%) | 87% (82%) | 89% (83%) | 89% (84%) | 88% (86%) | 89% (86%) | HD ([8, 32, 33]) |
| Visitor compliance with hand hygiene (effectiveness) | 61% (81%) | 62% (78%) | 63% (80%) | 76% (80%) | 81% (82%) | 82% (82%) | [8, 23, 24, 28] ([8, 32]) |
| Patient compliance with hand hygiene (effectiveness) | 58% (80%) | 59% (78%) | 60% (80%) | 60% (79%) | 60% (82%) | 61% (82%) | [29–31] ([8, 32]) |
| Contact Precautions Parameters | | | | | | | |
| Healthcare workers compliance with contact precautions (effectiveness) | 82% (68%) | 83% (66%) | 84% (67%) | 88% (67%) | 90% (69%) | 91% (69%) | [32] ([8, 34–37]) |
| Visitors compliance with contact precautions (effectiveness) | 61% (68%) | 62% (66%) | 63% (67%) | 66% (67%) | 66% (69%) | 67% (69%) | [32] ([8, 34–37]) |
| Environmental Cleaning Parameters | | | | | | | |
| Standard Patient | | | | | | | |
| Daily cleaning compliance | 56% (44%) | 57% (42%) | 58% (43%) | 73% (43%) | 79% (45%) | 79% (44%) | [38–43] ([8]) |
| Terminal cleaning compliance (effectiveness) | 56% (98%) | 57% (95%) | 58% (97%) | 90% (100%) | 98% (100%) | 98% (100%) | [38, 44–47] ([8, 48]) |
| CDI Patient | | | | | | | |
| Daily cleaning compliance (effectiveness) | 56% (96%) | 57% (94%) | 58% (96%) | 74% (96%) | 79% (99%) | 79% (98%) | [38–43] ([8, 48]) |
| Terminal cleaning compliance (effectiveness) | 57% (100%) | 57% (100%) | 59% (100%) | 92% (100%) | 98% (100%) | 98% (100%) | [38, 44–47] ([8, 48]) |
| Admission Parameters | | | | | | | |
| Proportion of patients who are colonized asymptomatically at admission | 6.10% | | | | | | [8] |
| Proportion of patients with CDI at admission | 0.29% | | | | | | [8] |
| Patient arrival rate | 72/day | | | | | | HD |
| CDI Testing Algorithm Parameters | | | | | | | |
| Percent of patients with unexplained gastrointestinal symptoms at admission | 2% | | | | | | HD |
| Percent of patients with possible non-CDI causes of gastrointestinal symptoms after 48 hours | 43% | | | | | | HD |

HD = Hospital data

() = Effectiveness parameter estimate

and water or alcohol-based hand rub. We used these counts and the estimated effectiveness of soap and water and alcohol-based hand rub to estimate the overall effectiveness of each hand hygiene event in our model (see Section B in S1 Appendix for additional data descriptions). Because patient and visitor hand hygiene data were not available, we scaled the baseline compliance and effectiveness estimates according to observed healthcare worker hand hygiene

compliance and effectiveness. Baseline estimates were derived from the generic model and more recent studies. Parameters for daily and terminal environmental cleaning compliance and effectiveness were also scaled similarly. Sections B, C, and E in S1 Appendix provide additional details related to parameter estimation and scaling. Parameters that could be directly estimated from hospital administrative data, such as the admission rate and testing algorithm parameters, were not subject to this scaling process.

## Validation experiments for replicating observed CDI data over time

To test the ability of our model to replicate CDI trends in the target hospital, we ran 5000 replications of the H-ABM for the years 2013–2018 and compared the results to actual HA-CDI data for each year. We used 5000 replications to generate stable outcome estimates with narrow 95% confidence intervals. CDI rates were recorded as the number of hospital-associated cases per 10,000 patient days, consistent with the CDC's standard definition for HA-CDI [49].

## Validation of socio-environmental network structure via colonization pressure

Colonization pressure is a measure of infectious pathogen prevalence in a susceptible individual's environment. Multiple studies have already found that patients residing in wards with high CDI prevalence, and therefore high colonization pressure, have increased risk of developing CDI [50, 51]. Theoretically, this leads to increased opportunities for *C. difficile* propagation, as more members of the socio-environmental network are likely to be contaminated. For example, healthcare workers within a high colonization pressure socio-environmental network (such as a single hospital ward), would be more likely to become contaminated with *C. difficile*. Therefore, they can more easily propagate *C. difficile* to non-infected patients, thereby increasing the risk of developing CDI for each patient in their network. Colonization pressure is an example of emergent behavior in the H-ABM, and does not directly determine if a susceptible patient develops CDI during a hospital stay [52]. Thus, an association between colonization pressure and risk of developing CDI could serve as a proxy to assess if the socio-environmental networks in H-ABM are reflective of reality. To validate the modeled socio-environmental networks using colonization pressure, we used the following equation adapted from the study by Dubberke et al. (2007) to calculate the mean colonization pressure (MCP) for each patient agent in every year of the historical simulation [50].

$$\text{MCP} = \frac{\sum_{\text{LOS}}\text{Number of colonized or infected patients in ward, per day}}{\text{Total number of susceptible days}}$$

For each day that a patient was susceptible, the numerator was increased by the number of colonized or infected patients in the same ward. Patients that were known to be colonized or infected with *C. difficile* at some point in their stay were classified as "case" patients. Community-associated colonizations or infections were not considered as case patients. However, community-associated colonizations or infections contributed to the number of colonized or infected patients in their respective wards per day. Any susceptible patients that did not become colonized or infected were considered "non-case."

## Impact of infection control interventions on CDI rates in the generic hospital model vs H-ABM

We estimated the effect of hospital-specific characteristics of H-ABM by comparing its impact on HA-CDI with that of the generic hospital ABM (as described in Barker, et al. [8]) under

similar conditions. We considered three discrete levels of infection control implementation: "baseline," "enhanced," and "ideal" corresponding to increasing levels of compliance and effectiveness parameters. We ran 5000 replications of the H-ABM with a single infection control intervention implemented with the "enhanced" or "ideal" parameters from Barker, et al., with all others implemented at "baseline." To assess the impact of each intervention, we also ran 5000 replications of H-ABM with all interventions' "baseline" parameters. For each experiment we collected the rate of HA-CDIs per 10,000 patient days, as well as the rate of asymptomatic colonizations per 1,000 admissions. Asymptomatic *C. difficile* colonization rates are considered of interest to hospital infection control, as an asymptomatically colonized patient may still propagate infectious spores [53]. Finally, we ranked each of the eight interventions according to their reduction in HA-CDI for both the H-ABM and generic model.

### Statistical and sensitivity analysis

For all experiments in this study, we calculated 95% confidence intervals with the SciPy stats package for the Python programming language [54]. We conducted a sensitivity analysis on three parameters of interest: the proportion of patients colonized at admission, the proportion infected at admission, and the CDI transition probability matrix.

### Ethics declaration

This study was approved by the target hospital's Institutional Review Board.

## Results

Estimates for model input parameters for 2013–2018 are shown in Table 1. Hospital reported rates of observed healthcare worker hand hygiene compliance were over 80% in all years. To account for this unrealistically high compliance, we assumed that these observed rates of hand hygiene represented the compliance rates when interacting with known CDI patients. The target hospital reported rates of healthcare worker soap and water, or alcohol-based hand rub usage were consistent with literature.

Using the estimated hospital and intervention parameters, the model was able to reproduce CDI trends from 2013–2018. Fig 3 shows the rate of HA-CDI per 10,000 patient days as predicted by the simulation with 95% confidence intervals for each yearly estimate and the actual rates by year. Notably, our results show a 46% percent drop in simulated HA-CDI per 10,000 patient days from 2015 to 2016 (13.47 to 7.23), akin to the 46% drop in the historical rate (14.01 to 7.58).

Across all simulated years, the average total number of case and non-case patients was approximately 687 and 91,283, respectively. The average MCP for all patients and patients without CDI was approximately 0.36. For patients with CDI, the average MCP was approximately 0.42. To further analyze the effect size of colonization pressure, we split the case and non-case population into three subgroups of MCP based on the case and non-case population averages: <0.36, 0.36–0.42, and > 0.42. Approximately 36% of case patients belong to the highest MCP group (> 0.42), vs 30% of non-case patients. Similarly, approximately 61% of case patients belonged to the lowest MCP group (< 0.36), vs 68% of non-case patients. In the H-ABM, greater than average MCP (0.36) was associated with a 37% increase in risk of CDI colonization or infection (Risk ratio: 1.37; 95% CI: [1.17, 1.59]). Analysis of the MCP groups demonstrated that increased MCP was associated with a significant risk of CDI colonization or infection, with patients in higher-than-average MCP groups having up to 66% increased risk of CDI (Table 2).

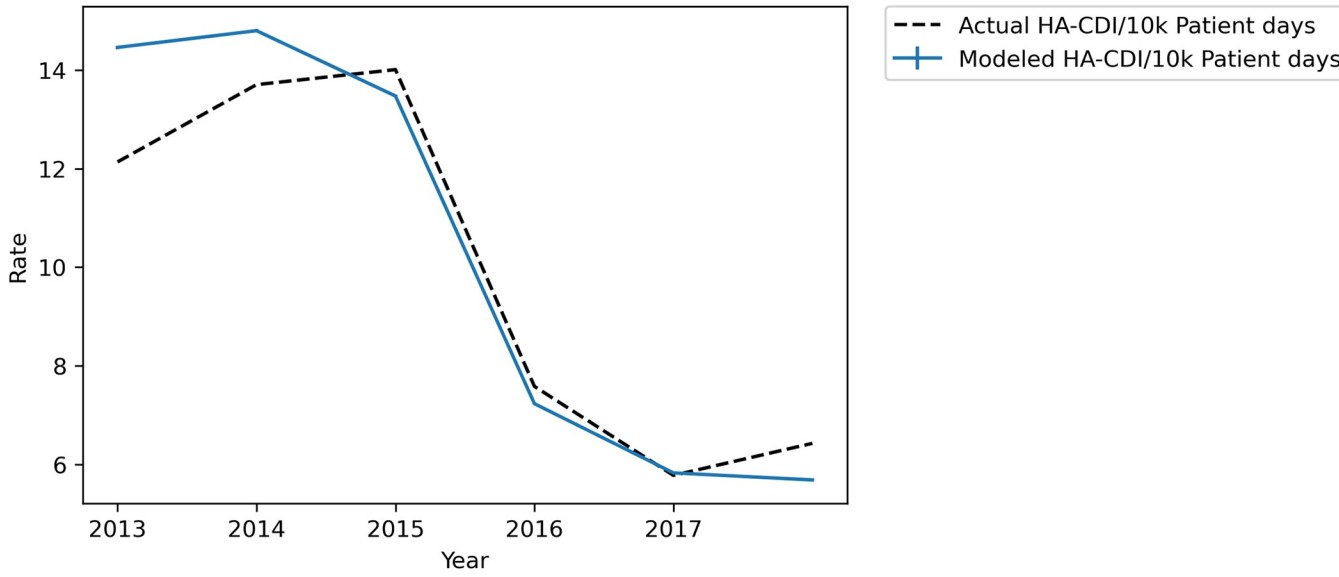

**Fig 3. Modeled vs. actual rate of HA-CDI per 10,000 patient days, 2013–2018.**

Table 3 displays the changes in HA-CDI and asymptomatic colonization rates relative to the baseline implementation for both the H-ABM and the generic model and ranks each of the eight infection control interventions by reduction in HA-CDI rate and asymptomatic colonizations. The baseline rate of HA-CDI per 10,000 patient days was 8.28 and 20.10 for the generic model and H-ABM, respectively. In the experiments conducted by both models, daily cleaning was consistently associated with the largest reduction in HA-CDI and colonization rates. However, screening at admission is less effective when implemented at the enhanced level in the H-ABM than in the generic model, and is associated with a smaller reduction in HA-CDI rates than enhanced healthcare worker hand hygiene. Full HA-CDI and asymptomatic colonization rates by intervention are included in Section F of S1 Appendix.

## Discussion

The H-ABM was able to replicate observed HA-CDI trends from 2013–2018, including the large relative drop in HA-CDI from 2015 to 2016 that most likely resulted from increased use of infection control interventions. Point estimate discrepancies could be due to historical fluctuations in the community prevalence of CDI, in adherence to/effectiveness of infection controls, and in the number of susceptible patients, among several other causes. In particular, the increase in historical HA-CDI rates from 2013–2015 could be the result of many difficult to detect hospital characteristics. One possibility is the transition from institutional definitions of HA-CDI to CDC definitions, and gradual changes in the reporting structure of CDI starting around the year 2012 [55]. Most of these causes are nearly impossible to capture via an ABM

**Table 2. Relative risk of CDI by MCP.**

| Mean Colonization Pressure (MCP) Group | Average Number of Case Patients | Average Number of Non-case Patients | Risk Ratio (95% CI) |
|---|---|---|---|
| < 0.36 | 420.7 | 62411.5 | Reference |
| > 0.36 | 266.5 | 28871.7 | 1.37 (1.17, 1.59) |
| 0.36–0.42 | 20.1 | 1784.4 | 1.66 (1.07, 2.60) |
| > 0.42 | 246.4 | 27087.3 | 1.35 (1.17, 1.60) |

**Table 3. Ranking of infection control interventions (1–8) by percent reduction in HA-CDI per 10,000 patient days, for the H-ABM and the generic model; with 1 representing the greatest reduction and 8 representing the least reduction.**

| H-ABM | | | | Generic Model | | |
|---|---|---|---|---|---|---|
| Intervention | HA-CDI per 10k patient days | | | Intervention | HA-CDI per 10k patient days | |
| | Enhanced | Ideal | | | Enhanced | Ideal |
| Daily cleaning | 65% (1) | 72% (1) | | Daily cleaning | 69% (1) | 72% (1) |
| Healthcare worker contact precautions | 2% (6) | 3% (6) | | Healthcare worker contact precautions | 1% (6) | 3% (6) |
| Healthcare worker hand hygiene | 30% (2) | 49% (2) | | Healthcare worker hand hygiene | 33% (3) | 52% (2) |
| Patient hand hygiene | 11% (4) | 21% (3) | | Patient hand hygiene | 12% (5) | 22% (5) |
| Screening at admission | 18% (3) | 19% (4) | | Screening at admission | 36% (2) | 37% (3) |
| Terminal cleaning | 8% (5) | 11% (5) | | Terminal cleaning | 18% (4) | 24% (4) |
| Visitor hand hygiene | 0% (7) | 0% (7) | | Visitor hand hygiene | 0% (7) | 0% (7) |
| Visitor contact precautions | 0% (7) | 0% (7) | | Visitor contact precautions | 0% (7) | 0% (7) |

and attempting to capture all of them may result in "overfitting," reducing the utility of our model for planning future infection control strategies. To avoid this "overfitting" we focused primarily on representing CDI trends and infection control relative performance. We believe our results provide a useful illustration of the model's ability to assess the relative impact of different infection control interventions used simultaneously.

This study also presents an alternate to common validation methods that rely on the availability of large quantities of outcomes data. Common validation approaches use calibration to estimate unknown parameters and reconcile modeled and actual outcomes of interest. While such methods may be appropriate for systems with vast historical data, this is not feasible for the hospital-specific setting. To overcome this, our validation approach emphasizes incorporating relevant system-specific features, such as physical layout, intervention adherence, and by-ward susceptibility, into the logic of the model.

Our colonization pressure analysis agrees with the existing empirical studies that demonstrate a significant effect size of colonization pressure on risk of developing CDI [50, 51]. However, our risk ratios for the effect size of MCP differ from literature reporting unadjusted risk ratios ranging from close to 1 to 8.7 for varying levels of colonization pressure [50, 51]. The discrepancy of our risk ratios ranging from 1.35–1.66 with those found in literature could be the result of many hospital-related factors. Lower adherence to environmental cleaning for rooms housing CDI patients may lead to increased spore prevalence in wards, resulting in more CDI cases and higher impact of colonization pressure. Other hospital-specific factors, such as the number of double occupant rooms, size of wards, and the distribution susceptible patients and of healthcare workers may have impacted the effect size of MCP in other studies. Our MCP effect size discrepancy may also be the result of simulation-specific factors. In our simulation, we assume that once a patient has been tested, their CDI status is known by all HCWs and patients in the model. In an actual hospital setting, patient CDI status may not always be known ubiquitously, resulting in insufficient infection control practices and an increased effect of colonization pressure as *C. difficile* spore prevalence increases. Additionally, our H-ABM follows strict testing rules independent of known CDI prevalence. Testing in a real hospital may be biased, as clinicians may be more likely to test patients for CDI if another known CDI patient is housed in the same ward. The effect size of colonization pressure is relatively understudied for CDI, but studies examining comparable infections like Methicillin-resistant *Staphylococcus aureus* and Vancomycin-resistant *Enterococcus* have found a more similar effect size for the highest colonization pressure group as our H-ABM [56–58].

Experiments examining the single intervention reduction in HA-CDI and asymptomatic colonization rates in the H-ABM and generic model demonstrate the utility of hospital

-specific modeling. The change in the relative ranking of high-impact interventions when adapting a generic model to a specific setting suggests that generic models are limited in their applicability to real-world settings. For the many hospitals that need to create infection control strategies under tight budget constraints, the relative ranking of different interventions is especially salient information. As ABMs become increasingly popular in healthcare applications, it is of growing importance to understand the tradeoffs between types of models. While generic models may be simpler to create, they may not capture the nonlinearities that arise from hospital-specific characteristics. Conversely, hospital-specific models may facilitate greater confidence in decision makers, but are more data-intensive and the incremental benefit of hospital-specific model findings against generic model findings is not always clear. The literature on hospital ABMs reveals that both generic and hospital-specific models can produce valuable insights [4, 59, 60]. Our study adds to this existing literature by demonstrating the utility of both generic and hospital-specific modeling. We hope our results may be of use to other modelers as they consider the tradeoff between the flexibility of a generic model and tailored findings of a hospital-specific model. To the best of our knowledge, our experiments are the first to attempt to directly characterize this tradeoff by analyzing two analogous models, therefore direct comparison to similar studies is limited.

Our H-ABM validation has several limitations. Compliance and effectiveness for environmental cleaning and contact precautions were not directly derived from hospital data and were instead scaled from other intervention data. Extensive data collection efforts would have been consistently required from 2013–2018 to accurately estimate these parameters from hospital data. However, this level of data collection is often infeasible in hospitals. Reliable intervention data would also need to be obtained covertly, thereby increasing data collection costs [61, 62]. The lack of extensive historical data regarding patient and visitor behavior further limits the generalization of study and H-ABM findings to the target hospital setting itself. Thus, it is necessary to develop more novel methods of estimating unknown parameters using proxies. Our study also does not include any calibration of model parameters, which is uncommon among ABM validation studies. Previously published ABMs of HA-CDI calibrated only the transition probability matrix that governs CDI progression using a large data set and estimates derived from literature [5]. Any additional calibration effort may not be appropriate for this study because the impact of hospital configuration primarily affects the rate at which susceptible patients are exposed to *C. difficile;* an event fully determined by model assumptions and logic. However, sensitivity analyses show that the overall CDI trends from 2013–2018 produced by the H-ABM are consistent across different hospital assumptions, as seen in Section G of S1 Appendix. By focusing on representing historical conditions to the best of our ability, we may consider this study as a type of external validation of the model's logic and assumptions. The performance of the external validation suggests the logic framework of H-ABM could be applicable to other healthcare settings. Finally, we acknowledge that MCP is an incomplete metric for validating the socio-environmental networks of an ABM. Ideally, the use of MCP as a validation metric would complement a validation that tracks CDI spread empirically through networks. However, given the lack of data available for such a thorough study, we believe that MCP provides a suitable aggregation of the socio-environmental networks to qualitatively demonstrate the fidelity of modeled networks. To the best of our knowledge, this study is the first to use MCP to support validation, and we hope that future work will expand and improve its usage for this purpose.

Further work is necessary to continue validation of the H-ABM. Comprehensive data for all infection control intervention implementation should be collected for several years with the goal of model validation. Patient heterogeneity and risk factors for CDI, aside from age, admission ward, and antibiotic use, should also be incorporated into the model. To strengthen the

validation of social and environmental networks in the model, a supplementary study is needed to determine the effect size of colonization pressure in the target hospital.

## Conclusion

In conclusion, the H-ABM was able to replicate CDI trends in the target hospital using parameters estimated from a combination of primary hospital data and literature. Through this validation process and experiments, we demonstrate some of the tradeoffs between generic and hospital-specific modeling. Despite the need for additional work, this study provides an alternate to common validation methods.

## Supporting information

**S1 Appendix. Supplementary material.**
(DOCX)

## Author Contributions

**Conceptualization:** Nasia Safdar, Oguzhan Alagoz.

**Formal analysis:** Elizabeth Scaria.

**Funding acquisition:** Nasia Safdar.

**Investigation:** Elizabeth Scaria.

**Methodology:** Elizabeth Scaria, Nasia Safdar, Oguzhan Alagoz.

**Project administration:** Oguzhan Alagoz.

**Resources:** Nasia Safdar, Oguzhan Alagoz.

**Software:** Elizabeth Scaria.

**Supervision:** Nasia Safdar, Oguzhan Alagoz.

**Validation:** Oguzhan Alagoz.

**Visualization:** Elizabeth Scaria.

**Writing – original draft:** Elizabeth Scaria, Oguzhan Alagoz.

**Writing – review & editing:** Elizabeth Scaria, Nasia Safdar, Oguzhan Alagoz.

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
