## [Decision Letter · Decision Letter 0]

22 Dec 2022

PONE-D-22-33383Validating agent-based simulation model of hospital-associated Clostridioides difficile infection using primary hospital dataPLOS ONE

Dear Dr. Alagoz,

Thank you for submitting your manuscript to PLOS ONE. After careful consideration, we feel that it has merit but does not fully meet PLOS ONE’s publication criteria as it currently stands. Therefore, we invite you to submit a revised version of the manuscript that addresses the points raised during the review process.

We look forward to receiving your revised manuscript.

Kind regards,

Mabel Kamweli Aworh, DVM, MPH, PhD. FCVSN

Academic Editor

PLOS ONE

Journal Requirements:

"I have read the journal's policy and the authors of this manuscript have the following competing interests: Oguzhan Alagoz has served as a consultant to Johnson & Johnson, Bristol Myers Squibb, Exact Sciences, and Biovector Inc. all of which are outside of the submitted work. Other authors report no conflict of interest. "

Additional Editor Comments:

1. The authors should highlight the limitations of this study.

Reviewers' comments:

Reviewer's Responses to Questions

**Comments to the Author**

1. Is the manuscript technically sound, and do the data support the conclusions?

Reviewer #1: Yes

Reviewer #2: Yes

2. Has the statistical analysis been performed appropriately and rigorously? 

Reviewer #1: Yes

Reviewer #2: Yes

3. Have the authors made all data underlying the findings in their manuscript fully available?

Reviewer #1: No

Reviewer #2: Yes

4. Is the manuscript presented in an intelligible fashion and written in standard English?

Reviewer #1: Yes

Reviewer #2: Yes

5. Review Comments to the Author

Reviewer #1: My overall impression is that this is a novel study where the authors tried to validate an already established model for infectious disease. This is critical to building trust in the output of these models given their utility in decision making by infection prevention and control experts within health facilities and in outbreak situations. The research attempts to circumvent the issues of extensive data requirements of the generic ABM while validating it. The abstract and introduction were well written. There is however the need to elaborate further on the burden of Clostridiodes difficile in terms of its prevalence as a HAI and the significance of its morbidity and mortality in order to put the relevance of this study in context.

The lack of data on patients and visitors IPC behaviour for use within the model limits the generalization of the findings of this study within the hospital setting. the authors to include the ethical review process for this work. Table 1 is confusing, the authors need to provide an explanation for the two percentages within each cells. The discussion provides an explanation for the findings but the research team have not sufficiently discussed their findings in the context of similar published works. The conclusions are in line with the objectives of the study and show the use of the H-ABM as an external validation model for the generic ABM and the significant utility derived from conducting a hospital specific ABM.

I personally think this is a well written manuscript, the findings are evidence that should be carefully considered by epidemiologists and IPC experts. I recommend that the manuscript be published after the raised queries within this review have been addressed.

Reviewer #2: The manuscript has been clearly written and the points succinctly presented. It is an original work of the author and is commendable and adds to the body of knowledge, however there a few minor corrections that need to be addressed.

6. PLOS authors have the option to publish the peer review history of their article (what does this mean?). If published, this will include your full peer review and any attached files.

Reviewer #1: **Yes: **Jenny A. Momoh

Reviewer #2: **Yes: **Rahab Charles-Amaza

---

## [Author Response · Author response to Decision Letter 0]

29 Jan 2023

Additional Editor Comments

0.1 The authors should highlight the limitations of this study.

Response: We thank the editor for their consideration of our manuscript. We have included a discussion of our study’s limitations in paragraph 5 of the Discussion section. We have enhanced this discussion of study limitations according to reviewer comment 2.3 below.

Reviewer #1

1.1 My overall impression is that this is a novel study where the authors tried to validate an already established model for infectious disease. This is critical to building trust in the output of these models given their utility in decision making by infection prevention and control experts within health facilities and in outbreak situations. The research attempts to circumvent the issues of extensive data requirements of the generic ABM while validating it. The abstract and introduction were well written. 

Response: We thank the reviewer for their thoughtful feedback, which we have used to strengthen our manuscript greatly.

1.2 There is however the need to elaborate further on the burden of Clostridiodes difficile in terms of its prevalence as a HAI and the significance of its morbidity and mortality in order to put the relevance of this study in context.

Response: To affirm the importance of healthcare-associated CDI prevention, we have added the following text to the fourth paragraph of the Introductions.

“CDI is one of the most common hospital associated infectious diseases ion the US, with up to 500,000 cases and 29,000 attributed deaths per year, and its control has long been a national public health priority [1]. ”

1.3 The lack of data on patients and visitors IPC behaviour for use within the model limits the generalization of the findings of this study within the hospital setting. 

Response: To acknowledge this limitation of our study, we have added the following text to paragraph 5 of the Discussion section of our manuscript: “The lack of extensive historical data regarding patient and visitor behavior further limits the generalization of study and H-ABM findings to the target hospital setting itself.”

1.4 the authors to include the ethical review process for this work. 

Response: We have included an ‘Ethics Declaration’ subsection at the end of the Methods section. Under this new section, we state: “This study was approved by the target hospital’s Institutional Review Board.”

1.5 Table 1 is confusing, the authors need to provide an explanation for the two percentages within each cells. 

Response: We apologize for the lack of clarity in the formatting for Table 1 in the original submission. The percent values in parentheses represent the effectiveness of each of the interventions. To clarify this distinction, we have added ‘(effectiveness)’ to the first column of each applicable row, to denote that the effectiveness parameter is included within parentheses. Additionally, we have added ‘( ) = Effectiveness parameter estimate’ at the foot of the table, to clarify the formatting decision.

1.6 The discussion provides an explanation for the findings but the research team have not sufficiently discussed their findings in the context of similar published works. The conclusions are in line with the objectives of the study and show the use of the H-ABM as an external validation model for the generic ABM and the significant utility derived from conducting a hospital specific ABM.

Response: To better contextualize the findings expressed in our manuscript, we have added the following text to paragraph 4 of the Discussion section (new text italicized).

“The literature on hospital ABMs reveals that both generic and hospital-specific models can produce valuable insights [2–4]. Our study adds to this existing literature by demonstrating the utility of both generic and hospital-specific modeling. We hope our results may be of use to other modelers as they consider the tradeoff between the flexibility of a generic model and tailored findings of a hospital-specific model. To the best of our knowledge, our experiments are the first to attempt to directly characterize this tradeoff by analyzing two analogous models, therefore direct comparison to similar studies is limited.”

1.7 I personally think this is a well written manuscript, the findings are evidence that should be carefully considered by epidemiologists and IPC experts. I recommend that the manuscript be published after the raised queries within this review have been addressed.

Response: We thank the reviewer again for their thoughtful feedback and support of our manuscript.

Reviewer #2 

2.1 The manuscript has been clearly written and the points succinctly presented. It is an original work of the author and is commendable and adds to the body of knowledge, however there a few minor corrections that need to be addressed.

Response: We thank the reviewer for their consideration and their constructive comments. 

2.2 General comments 

This is an important study with interesting findings. It is quite commendable and will add to the body of knowledge.

Response: We thank the reviewer for their support of our manuscript.

2.3 Have a uniform in-text referencing according to the journal guideline. Sometimes, the references come after the period and other times, the references appear before the period.

Response: We have reviewed our manuscript document and have standardized all in-text references.

2.4 Abstract

The abstract should take an IMRAD structure (Introduction, Methods, Results and Discussion) and there is also no need for the headings to be spelt out into the various components, it should just assume a flow.

Response: We have removed the headings and made minor adjustments within the Abstract to present all components of the IMRAD structure in a single flow.

Introduction

2.5 Line 7- ISPOR-SMDM should be spelt out in full for future references

Response: We have included this change in the revised manuscript.

2.6 Line 16 to 31– Should be referenced

Response: In this section, we express the difficulty of validating ABMs of hospitals given the lack of primary data. Hospitals typically lack this data because of the high cost of data collection, or because some data is not directly observable. Following the referee’s recommendation, we now added a few citations to support our claim. This section is included below for reference (inserted text italicized, deleted text struck through):

“Challenges associated with validation posed by insufficient data are amplified when modeling specific hospitals because primary hospital data can be particularly scarce given the high cost of data collection. This lack of data is exemplified by the number of hospital ABMs that leverage at least some mix of primary hospital data and literature or secondary data sources [5–9]. Therefore, approaches to validating ABMs of specific healthcare settings likely need to incorporate both primary data as well as data from literature. To the best of our knowledge, there is currently a dearth of formalized procedures that explore possibilities for blending literature and primary data sources to contribute to model validation. 

Additionally, ABMs depend heavily on the networks facilitating interactions between agents and their environment, yet validating socio-environmental networks is especially challenging [10]. For example, ABMs may record the frequency of interactions among agents in a system (e.g., patients to patients, or patients to healthcare workers) that contribute to the transmission of a disease, but there is paucity of data and methods to validate these frequencies and the overall socio-environmental network structure. As a result, modelers might need to consider indirect metrics of network behavior to validate socio-environmental networks, such as emergent network effects, but to the best of our knowledge this approach has not been employed in the context of modeling disease transmission in a hospital setting.”

2.7 Line 35- Clostridium difficile infection should be term CDI not (C. difficile; CDI)

Response: We have included this change in the revised manuscript.

2.8 Line 37- Change “adapt’’ to adapted

Response: We have included this change in the revised manuscript.

2.9 Line 43- Change ‘’compare’’ to compared

Response: We have included this change in the revised manuscript.

2.10 Methods - Change ‘’CDI Background’’ to Background or introduction

Response: We have included this change in the revised manuscript.

2.11 Overview of the existing agent-based simulation model developed for a generic hospital setting: Line 7 to 22 – It is too lengthy, should be rephrased and properly referenced

Response: We have shortened this section by using more concise language, and we have added citations for additional claims made within this section. 

2.12 Modeling infection control interventions: Line 5- Change ‘’define’’ to defined

Response: We have included this change in the revised manuscript.

2.13 Validation of socio-environmental network structure via colonization pressure: Line 5 to 13- Should be referenced 

Response: In this section, we propose and argue for the use of colonization pressure as a proxy to validate socio-environmental networks in the H-ABM. To support our argument, we describe how colonization pressure is linked to the socio-environmental networks within a hospital. To clarify this purpose, we have rewritten parts of this section, and added citations for any claims. This section is included below for reference (inserted text italicized, deleted text struck through):

“For example, healthcare workers within a high colonization pressure socio-environmental network (such as a single hospital ward), would be more likely to become contaminated with C. difficile. Therefore, they can more easily propagate C. difficile to non-infected patients, thereby increasing the risk of developing CDI for each patient in their network. Colonization pressure is an example of emergent behavior in the H-ABM, and does not directly determine if a susceptible patient develops CDI during a hospital stay [11]. Thus, an association between colonization pressure and risk of developing CDI serves could serve as a proxy to assess if the socio-environmental networks in H-ABM are reflective of reality. To validate the modeled socio-environmental networks using colonization pressure, we used the following equation adapted from the study by Dubberke et al. (2007) to calculate the mean colonization pressure (MCP) for each patient agent in every year of the historical simulation [12].”

2.14 Line 12- Grammatical error should be addressed

Response: We have fixed this grammatical error in the revised manuscript.

Discussion

2.15 In a s much as there was detailed discussion which included the conclusion, the conclusion should be separated from it to stand on its own. 

Response: To emphasize the conclusions of this study, we have created a Conclusions section following the Discussion section.

2.16 Line 22- Paraphrasing and referencing will be preferred rather than using “the existing literature’’ and referencing the literature, kindly effect changes.

Response: Following the referee’s recommendation, we have restructured this line to cite the specific studies and paraphrased the primary findings of these studies (i.e., that colonization pressure has a significant effect size on a patient’s risk of CDI).

2.17 List of Abbreviations

Several acronyms were used in your manuscript hence a list of abbreviations used should be made available.

Response: We have included a List of Abbreviation after the Conclusions of the revised manuscript.

Journal Requirements

3.1 Please ensure that your manuscript meets PLOS ONE's style requirements, including those for file naming. The PLOS ONE style templates can be found at 

Response: We have reviewed our manuscript formatting and style and can confirm that our manuscript adheres to these requirements.

3.2 Thank you for stating the following in the Competing Interests section: 

"I have read the journal's policy and the authors of this manuscript have the following competing interests: Oguzhan Alagoz has served as a consultant to Johnson & Johnson, Bristol Myers Squibb, Exact Sciences, and Biovector Inc. all of which are outside of the submitted work. Other authors report no conflict of interest. "

Response: We have included an amended Competing Interests statement in our cover letter for this revision. For reference, the amended statement is as follows: I have read the journal's policy and the authors of this manuscript have the following competing interests: Oguzhan Alagoz has served as a consultant to Johnson & Johnson, Bristol Myers Squibb, Exact Sciences, and Biovector Inc. all of which are outside of the submitted work. Other authors report no conflict of interest. This does not alter our adherence to PLOS ONE policies on sharing data and materials.”

3.3 We note that you have stated that you will provide repository information for your data at acceptance. Should your manuscript be accepted for publication, we will hold it until you provide the relevant accession numbers or DOIs necessary to access your data. If you wish to make changes to your Data Availability statement, please describe these changes in your cover letter and we will update your Data Availability statement to reflect the information you provide.

Response: Thank you for this opportunity to clarify our Data Availability statement. We believe that all data that we are at liberty to disclose is available in the text of this manuscript or in the Supporting Information. Therefore, we believe our Data Availability statement should be amended to the following: “All relevant data are within the manuscript and its Supporting Information files.” This amended statement is also included in our revision cover letter.

3.4 Please include captions for your Supporting Information files at the end of your manuscript, and update any in-text citations to match accordingly. Please see our Supporting Information guidelines for more information: http://journals.plos.org/plosone/s/supporting-information. 

Response: We have included a Supporting Information caption section at the end of our revised manuscript. 

3.5 Please review your reference list to ensure that it is complete and correct. If you have cited papers that have been retracted, please include the rationale for doing so in the manuscript text, or remove these references and replace them with relevant current references. Any changes to the reference list should be mentioned in the rebuttal letter that accompanies your revised manuscript. If you need to cite a retracted article, indicate the article’s retracted status in the References list and also include a citation and full reference for the retraction notice.

Response: We have removed one item from our reference list, as we no longer believe that it should be classified with the group of studies it had been previously cited in. The removed reference list item is provided below:

Liu Z, Rexachs D, Epelde F, Luque E. An agent-based model for quantitatively analyzing and predicting the complex behavior of emergency departments. J Comput Sci. 2017 Jul 1;21:11–23.

It had been previously referenced in the following text from paragraph 2 of the Introduction: “While several ABMs of healthcare settings exist, few have been validated using primary data from an extant system: often, validation efforts use community- or national-level historical data [2,6,13–15].” 

Other changes to the references list include new citations that were added according to reviewer comments. We confirm that all other references are complete and correct.

 

References

1. Guh AY, Mu Y, Winston LG, Johnston H, Olson D, Farley MM, et al. Trends in U.S. Burden of Clostridioides difficile Infection and Outcomes. N Engl J Med. 2020 Apr 1;382(14):1320–30. 

2. Barnes S, Golden B, Wasil E. MRSA Transmission Reduction Using Agent-Based Modeling and Simulation. Inf J Comput. 2010 Apr 15;22(4):635–46. 

3. Temime L, Pannet Y, Kardas L, Opatowski L, Guillemot D, Boëlle PY. NOSOSIM: an agent-based model of pathogen circulation in a hospital ward. In: Proceedings of the 2009 Spring Simulation Multiconference. San Diego, CA, USA: Society for Computer Simulation International; 2009. p. 1–8. (SpringSim ’09). 

4. Almagor J, Temkin E, Benenson I, Fallach N, Carmeli Y, on behalf of the DRIVE-AB consortium. The impact of antibiotic use on transmission of resistant bacteria in hospitals: Insights from an agent-based model. PLOS ONE. 2018 May 14;13(5):e0197111. 

5. Barker AK, Alagoz O, Safdar N. Interventions to reduce the incidence of hospital-onset Clostridium difficile infection: An agent-based modeling approach to evaluate clinical effectiveness in adult acute care hospitals. Clin Infect Dis Off Publ Infect Dis Soc Am. 2017 Nov 3; 

6. Codella J, Safdar N, Heffernan R, Alagoz O. An agent-based simulation model for Clostridium difficile infection control. Med Decis Mak Int J Soc Med Decis Mak. 2015 Feb;35(2):211–29. 

7. Stainsby H, Taboada M, Luque E. Towards an Agent-Based Simulation of Hospital Emergency Departments. In: 2009 IEEE International Conference on Services Computing. 2009. p. 536–9. 

8. Rubin MA, Jones M, Leecaster M, Khader K, Ray W, Huttner A, et al. A Simulation-Based Assessment of Strategies to Control Clostridium Difficile Transmission and Infection. PLOS ONE. 2013;8(11):e80671.

9. Barnes SL, Morgan DJ, Harris AD, Carling PC, Thom KA. Preventing the transmission of multidrug-resistant organisms (MDROs): Modeling the relative importance of hand hygiene and environmental cleaning interventions. Infect Control Hosp Epidemiol Off J Soc Hosp Epidemiol Am. 2014 Sep;35(9):1156–62. 

10. Macal CM, North MJ. Agent-based modeling and simulation. In: Proceedings of the 2009 Winter Simulation Conference (WSC). 2009. p. 86–98. 

11. Li Z, Sim CH, Hean Low MY. A Survey of Emergent Behavior and Its Impacts in Agent-based Systems. In: 2006 4th IEEE International Conference on Industrial Informatics. 2006. p. 1295–300. 

12. Dubberke ER, Reske KA, Olsen MA, McMullen KM, Mayfield JL, McDonald LC, et al. Evaluation of Clostridium difficile–Associated Disease Pressure as a Risk Factor for C difficile–Associated Disease. Arch Intern Med. 2007 May 28;167(10):1092–7. 

13. Taboada M, Cabrera E, Iglesias ML, Epelde F, Luque E. An Agent-Based Decision Support System for Hospitals Emergency Departments. Proc Int Conf Comput Sci ICCS 2011. 2011 Jan 1;4:1870–9. 

14. Liu Z, Rexachs D, Epelde F, Luque E. An agent-based model for quantitatively analyzing and predicting the complex behavior of emergency departments. J Comput Sci. 2017 Jul 1;21:11–23. 

15. Tracy M, Cerdá M, Keyes KM. Agent-Based Modeling in Public Health: Current Applications and Future Directions. Annu Rev Public Health. 2018 Apr 1;39(1):77–94.

---

## [Decision Letter · Decision Letter 1]

5 Apr 2023

Validating agent-based simulation model of hospital-associated Clostridioides difficile infection using primary hospital data

PONE-D-22-33383R1

Dear Dr. ALAGOZ,

We’re pleased to inform you that your manuscript has been judged scientifically suitable for publication and will be formally accepted for publication once it meets all outstanding technical requirements.

Kind regards,

Mabel Kamweli Aworh, DVM, MPH, PhD. FCVSN

Academic Editor

PLOS ONE

Reviewers' comments:

Reviewer's Responses to Questions

**Comments to the Author**

1. If the authors have adequately addressed your comments raised in a previous round of review and you feel that this manuscript is now acceptable for publication, you may indicate that here to bypass the “Comments to the Author” section, enter your conflict of interest statement in the “Confidential to Editor” section, and submit your "Accept" recommendation.

Reviewer #1: All comments have been addressed

Reviewer #2: All comments have been addressed

2. Is the manuscript technically sound, and do the data support the conclusions?

Reviewer #1: Yes

Reviewer #2: Yes

3. Has the statistical analysis been performed appropriately and rigorously? 

Reviewer #1: Yes

Reviewer #2: Yes

4. Have the authors made all data underlying the findings in their manuscript fully available?

Reviewer #1: No

Reviewer #2: Yes

5. Is the manuscript presented in an intelligible fashion and written in standard English?

Reviewer #1: Yes

Reviewer #2: Yes

6. Review Comments to the Author

Reviewer #1: Following the review of the updated manuscript, I think that the authors have to a very great extent addressed the queries raised. The burden of CDI has been expressed in terms of its morbidity and mortality although the prevalence of this issue was not stated. The authors have also included as one of its limitation, the lack of data on patient and visitors IPC behaviour. Table 1 has been properly formatted to allow for easy understanding and interpretation of the numbers in the cells. The discussion section has also been done with citation of more relevant literature.

I am satisfied with the current update on the manuscript as the authors have taken the time to address point by point all issues raised. I therefore recommend that the manuscript be published.

Reviewer #2: Yes, the manuscript was intelligently put together and presented and written in standard English that is easy to understand.

The manuscript is systematic and well put together, previous recommendations for corrections have been effected. However, in your discussion, it is always important to clearly state the health/public health implication of the work and how it has added to the body of knowledge.

7. PLOS authors have the option to publish the peer review history of their article (what does this mean?). If published, this will include your full peer review and any attached files.

Reviewer #1: **Yes: **Jenny A Momoh

Reviewer #2: **Yes: **Rahab Charles Amaza

---

## [Editor Report · Acceptance letter]

13 Apr 2023

PONE-D-22-33383R1 

Validating agent-based simulation model of hospital-associated *Clostridioides difficile* infection using primary hospital data 

Dear Dr. Alagoz:

I'm pleased to inform you that your manuscript has been deemed suitable for publication in PLOS ONE. Congratulations! Your manuscript is now with our production department. 

Kind regards, 

on behalf of

Dr. Mabel Kamweli Aworh 

Academic Editor

PLOS ONE